# Giving birth: A hermeneutic study of the expectations and experiences of healthy primigravid women in Switzerland

Valerie Fleming [1]*, Franziska Frank[2], Yvonne Meyer[3], Jessica Pehlke-Milde[4], Piroska Zsindely[4], Harriet Thorn-Cole[3], Claire de Labrusse[3]

1 Faculty of Health, Liverpool John Moores University, Liverpool, United Kingdom, 2 School of Sociology, University of Arizona, Tucson, Arizona, United States of America, 3 School of Midwifery, University of Health Sciences of Western Switzerland, Lausanne, Switzerland, 4 Research Unit for Midwifery Science, Zurich University of Applied Sciences, Winterthur, Switzerland

ᴑ These authors contributed equally to this work.
* v.fleming@ljmu.ac.uk

**Data Availability Statement:** Data are available from the University of Health Sciences (HESAV),

## Abstract

Switzerland experiences one of the highest caesarean section rates in Europe but it is unclear why and when the decision is made to perform a caesarean section. Many studies have examined from a medical and physiological point of view, but research from a women's standpoint is lacking. Our aim was to develop a model of the emerging expectations of giving birth and the subsequent experiences of healthy primigravid women, across four cantons in Switzerland. This longitudinal study included 30 primigravidae from the German speaking, 14 from the French speaking and 14 from the Italian speaking cantons who were purposively selected. Data were collected by semi-structured interviews taking place around 22 and 36 weeks of pregnancy and six weeks and six months postnatally. Following Gadamer's hermeneutic, which in this study comprised 5 stages, a model was developed. Four major themes emerged: *Decisions*, *Care*, *Influences* and *Emotions*. Their meandering paths and evolution demonstrate the complexity of the expectations and experiences of women becoming mothers. In this study, women's narrated mode of birth expectations did not foretell how they gave birth and their lived experiences. A hermeneutic discontinuity arises at the 6 week postnatal interview mark. This temporary gap illustrates the bridge between women's expectations of birth and their actual lived experiences, highlighting the importance of informed consent, parent education and ensuring women have a positive birth and immediate postnatal experiences. Other factors than women's preferences should be considered to explain the increasing caesarean section rates.

## Introduction

### Background

The central European country of Switzerland is constantly ranked amongst the highest of the world's wealthiest countries with residents enjoying a high quality of life [1]. In such contexts,

Lausanne, Switzerland Institutional Data Access / Ethics Committee. Researchers that meet the criteria for accessing the confidential data may contact Dr. Claire de Labrusse at claire. delabrusse@hesav.ch, or Veronika Schoeb at veronika.schoeb@hesav.ch.

**Funding:** The project was funded by Swiss National Science Foundation (Number 100017_147270). The funders had no role in study design, data collection and analysis, decision to publish, or preparation of the manuscript.

**Competing interests:** The authors have declared that no competing interests exist.

women expect safe and healthy birth outcomes for themselves and their babies [2]. Although there is a large number of places in which women can give birth, the healthcare system is generally highly medicalised and models of midwifery led care and continuity of care are rare [3]. Similarly, the caesarean section rate has steadily climbed in recent years, from 22.7% in 1998 to 32.0% in 2019 [4]. Recently a study on 18 EU countries showed that the Swiss rates of caesarean section *before the onset of labour* at after 37 weeks' gestation were among the highest (nulliparous 3.3%, multiparous 3% excluding previous CS) [5, 6].

Switzerland is not alone in this respect and the constantly increasing rate of caesarean sections in many industrialised countries is a hotly debated topic, in both public and professional fora. This is especially stressed as the World Health Organization (WHO) has consistently advocated for caesarean section rates to lie between 10 and 15% [7, 8]. This was confirmed in a systematic review and in an ecological analysis, suggested that there are no justifications for caesarean section rates above 15% in developed countries [9].

The specialist medical literature generally agrees that the rising caesarean section rate is due to medical indications. The most common medical indications for this medical intervention include suspected foetal compromise, foetal malpresentation, previous uterine surgery, multiple gestation, or slow progress [10, 11]. However, there is increasing evidence that the negative health consequences of caesarean section without a clear medical indication are underestimated, and that the higher rates do not improve maternal and neonatal mortality and morbidity rates [12]. Indeed a recent systematic review shows the reverse to be true [13] This dispels a widespread theory that the rise in caesarean sections can be attributed to an altered maternal risk profile. The large regional differences in caesarean section rates ranging from 19.4% to 40% between Swiss cantons calls for further questions [6].

Moreover, even in countries with high caesarean section rates, the majority of pregnant women prefer a vaginal birth and do not expect a caesarean section, unless there is a medical or obstetric need [14]. However, some women feel ambivalent at some point during their pregnancy, and those who have had a previous caesarean section elect to have a repeat one in their subsequent birth [15].

It remains unclear if research findings around short and long-term health consequences for mothers and babies born by caesarean sections, are made available to women in Switzerland, to assist them in making informed decisions about mode of birth. Until now, women's views have mainly been expressed in the media, with a focus on women's rights to autonomy in opting for a caesarean section. However, a few pregnant women, despite a more complex pregnancy, choose a birth setting that does not align with medical advice, notably as a result of a disagreement with their health care professional when planning the birth of their baby [16].

The topic has also become the frequent subject of discussions, concerning its necessity, accompanying costs and long-term effects on women's and children's health in Switzerland. Political interventions on the topic are taking place both at national and regional levels. Since the parliamentary discussion in 2013, mode of birth in Switzerland is a controversial topic amongst parliamentarians, and yet only minor political actions on regional levels were undertaken [17].

Despite the prevalent belief that many women choose to give birth by caesarean section, the limited research findings show evidence that only a very small number of women have this expectation. An explanation for this myth suggests the artificial maintenance of scientific controversy to justify the *status quo* of caesarean section practice, resulting in women's autonomous decision-making being impaired [18, 19]. Despite a growing body of research in this area, reflecting the importance of the topic, the number of high quality studies examining mode of birth in contexts in which women appear to have more of a choice, remains limited [20, 21]. Prior to becoming pregnant, women who have developed an interest and knowledge

around pregnancy and birth, and who do not fear birth, tend to expect to have a vaginal birth, whilst others would prefer a caesarean section [22]. Moreover, in countries with high caesarean section rates, the majority of studies agree that birth technologies make birth easier and women should have a right to choose a non-medically indicated caesarean [23].

Sumbul et al. 2020, through a review of the literature, demonstrate that most published studies show cross sectional pictures of women's expectations using predetermined questions [24]. Women's expectations of birth and how their life experiences influence them are particularly relevant issues for the present study. A study undertaken in Switzerland is of high relevance [25]. Although recruitment only took place after birth, the researchers investigated how 251 participants' views of their birth experiences changed in the first two years of their child's life. The study also sought to identify any particular groups of women at risk of developing a long-term negative memory of their birth experience. Women's birth experiences were collected and appraised within 48 to 96 hours postpartum, at three weeks and then in the second year after giving birth. The authors concluded that women at risk of developing a negative long-term memory of their birth experiences can be identified in the early postnatal period, when the overall birth experience and the perceived relationship to their intrapartum experience are considered. This study provides useful insights, but the varying parity of the participants and the lack of focus on their expectations leave some unanswered questions as to its validity.

Overall there appears to be a lack of clarity as to why and when the decision is made to undertake a caesarean section and which factors influence this process [26]. This, in turn, prompts the questions as to what expectations pregnant women have of their birth, how women make the decision for a particular mode birth, and how women recollect their experience of giving birth in relation to their decision making processes [27–29]. This study therefore focuses on healthy women becoming first time mothers, and their expectations and subsequent experiences of giving birth to provide a baseline understanding.

## Methodology

### Qualitative approach and theoretical framework

This study seeks to generate an understanding in women's journey to choosing a mode of birth and subsequently, how they experienced the birth of their baby. The hermeneutic of Gadamer is well suited for this study [30]. This philosophy aims to generate an in-depth understanding rather than simply describing an experience, and to draw out new knowledge from the participants. From Gadamer's philosophy, a specific five stage approach was adopted [31]. Table 1 present these stages and the fact that are not necessarily consecutive, but involve an engagement with the hermeneutic circle, going backwards and forwards between its whole and parts. Fig 1 shows the reflexive path we took.

Throughout this study, the research group members were aware of their own pre-understandings, gained through their own professional and personal experiences, and encounters [30]. These pre-understandings were recorded in the form of reflexive dialogues between team members and analysed alongside the acquired data. It was also acknowledged that these may change during the course of this study as shown in this statement, which emerged from a team discussion:

> "Well, midwives will have a different starting point from others such as me. We need to take all that into consideration when we do our analysis so that we are not letting professional biases interfere with our analysis"

**Table 1. Study's hermeneutic circle stages.**

| STAGE# | STAGE HEADING | STAGE DESCRIPTION |
|---|---|---|
| 1 | Generating the research question | The generation of the aim and research questions |
| 2 | Identifying pre-understandings | • Literature review<br>• Authors acknowledge of their interest and their own understandings of the topic through their professional experiences |
| 3 | Gaining understanding through dialogue with participants | • Participant selection (n = 58)<br>• Longitudinal semi-structured interviews (around 22 and 36 weeks antenatally (AN), and 6 weeks and 6 months postnatally (PN))<br>• Data analysis |
| 4 | Gaining understanding through dialogue with the texts | • Research group team members discuss the findings whilst taking into account the pre-understandings of stage 2. |
| 5 | Establishing trustworthiness | Generating the model by being true to the data |

## Research questions

To build the model, we sought to answer the following questions

- What are healthy primigravid women's expectations in early pregnancy about giving birth?

- How do these expectations change during pregnancy?

- What factors influence these expectations?

- What were the women's birthing experiences?

- How did the birthing experience match the antenatal expectations?

## Methods

### Setting and sampling strategy

Though a relatively small country, Switzerland comprises a heterogenous population. The country consists of a confederation of 26 cantons, and each canton has its own culture and

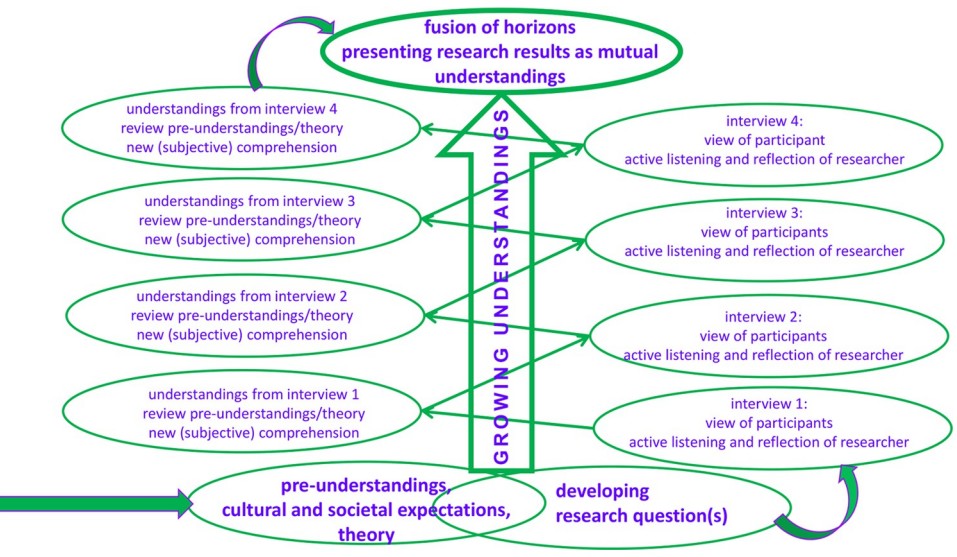

**Fig 1. The study's adapted hermeneutic circle [31].**

customs. German is spoken in the northern and eastern cantons, French in the western cantons, and Italian in the southern canton of Ticino. To reflect the geographical distribution of the country and the national birth statistics, we sampled 30 women from the German speaking, 14 from the French speaking and 14 from the Italian speaking cantons. We adopted a purposive sampling method, Table 2, appropriate for a hermeneutic study [32]. Our choice resonated with Gadamer's notion of cultural inclusiveness and maximising understanding of all elements of the data. It also took into account a minimum size required for generating data for each Swiss language [33].

Our sample was obtained from healthy primigravidae over the age of 18 with straightforward pregnancies. We recruited from settings which covered the full range of birth options from home birth to a university hospital, resulting in a wide range of potential participants, Table 2. Because of the chosen philosophical paradigm, the participants had to be fluent in their canton's official language (i.e. German, French or Italian). The 75 recruited women provided written consent in their chosen language.

## Data collection

We chose qualitative interviews as the most appropriate method for data collection. The interviewers were experienced, female researchers. Three were midwives, one a psychologist and one a sociologist. All were fluent speakers in the language in which they collected the data and in English. Participants were interviewed by the same allocated interviewer throughout the study.

We invited the 75 consenting women to participate in four guided interviews lasting approximately one hour at 20–24 weeks and 35–37 weeks antenatally, and six weeks and six months postnatally. Each interview began with the question "*What are your expectations of birth?*" or "*How was your birth in relation to your earlier expectations?*". Our longitudinal approach reflects the recommendations from reviews of both Gamble & Creedy [34] and McCourt et al. [35].

We collected data by semi-structured interviews at a place of each participant's choice. Interviews lasted between 45 and 75 minutes, were audio-recorded, transcribed verbatim and an initial thematic analysis was carried out before raising the themes to the next hermeneutic stage. During transcription all identifying details were removed and participants were given pseudonyms. MaxQDA© was used for data management and analysis. Transcripts were first entered in the interview's original language, before generating initial codes. Each researcher wrote memos in English pertaining to significant codes and to the interview in its entirety.

## Analysis

An initial thematic analysis of the data was undertaken following Braun and Clarke's proposed method [36]. In hermeneutic studies, this method allows to identify themes, in order to then draw out understandings of lived human experiences [37]. In addition, thematic analysis is a

**Table 2. Recruitment target numbers and actual numbers.**

| Recruitment setting | Zurich | St Gallen | Vaud | Ticino |
|---|---|---|---|---|
| Private obstetricians' practices | 6 (2) | 3 (0) | 3(2) | 3(2) |
| Public hospitals | 16 (10) | 8 (2) | 8 (8) | 8 (8) |
| Birth centres/independent midwives | 4 (4) | 2 (1) | 2 (2) | 2 (2) |
| Other | 4 (9) | 2 (2) | 2 (2) | 2 (2) |
| **Total** | **30 (25)** | **15 (5)** | **15 (14)** | **15 (14)** |

useful method to analyse large datasets. The interview memos formed the first point of discussion among the team as to commonalities and differences among participants within the same canton. The senior researchers on each site then compared the themes generated in each canton before undertaking an in-depth hermeneutic analysis, allowing for the understandings from combined themes from each language region to emerge. These were discussed by the complete team, whose members generated an initial model, which was revised on several occasions as the analysis deepened. The completed analysis permitted the identification of the key themes, illustrated using the translations of participants' own words. The themes and their longitudinal evolution in the antenatal and postnatal periods will form the elaborated model of "birth expectation to birth experience".

## Ethical considerations

The main ethical issues were informed consent, autonomy, confidentiality and anonymity. Primary permission to undertake the study was given by the Ethics Commission for Zürich (KEK-ZH-2014-0367). Secondary permission was granted by the ethics commissions of each of the other three cantons: Vaud, St Gallen and Ticino.

## Findings

Four main themes emerged from this longitudinal hermeneutic study: *Decisions*, *Care*, *Emotions*, and *Influences*. Each theme evolved from one interview to the next. At various stages of the pregnancy, the themes were either present, strongly present, or absent. Some merged with other themes, as detailed below in Fig 2.

## Decisions

In the first interviews, around 22 weeks antenatally (AN), women were experiencing a feeling of "being in limbo". At this particular stage of their pregnancy, women began to realise the enormity of the change that their pregnancy and arrival of their baby would bring to their lives:

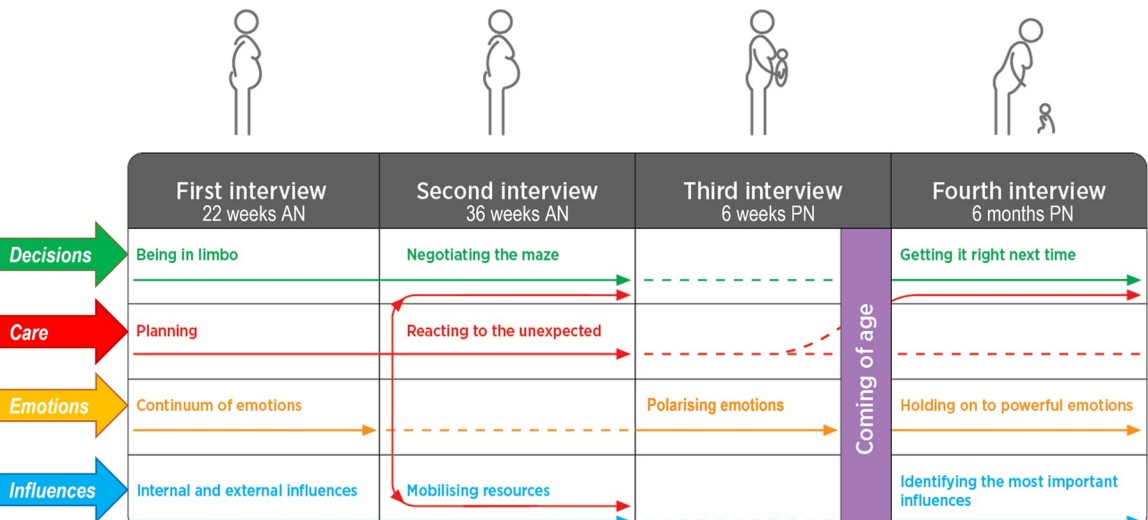

**Fig 2. Women's birth expectations and experiences: the evolution of the four themes along their pregnancy and postnatal journeys (AN = antenatal; PN: Postpartum).**

It's simply a new, yes new, new thing, something that I have never experienced before. [. . .] I think it's all the uncertainty that I don't like. [Ronja; 22 weeks AN].

For some women, this feeling led to a sense of denial:

During the first three months, I really. . .I forced myself not to build up any emotional relationship with this "egg". I called him "egg" at the beginning because I was really scared of having a miscarriage. [Scarlet; 22 weeks AN].

Others were more proactive and questioned their feelings:

I have started asking a lot of questions just about whatever I am feeling. . ...so far I have not really changed my mind but it's true that I have a lot of questions. [Emilia; 22 weeks AN]

By the time of the second interview, around 36 weeks, there was more of a sense of purpose, which we identified as "negotiating the maze". As Nora, who was seeking an elective caesarean for breech presentation, notes:

I'm more serene and less anxious thinking that it will be a caesarean, with a scheduled date. It's more reassuring for me, [. . .] I do not trust myself to give birth vaginally. [Nora; 36 weeks AN]

This theme was not present in the third interviews around 6 weeks postpartum but in the final interviews at 6 months after birth. After 6 months, women start focusing on the next pregnancy and "getting it right the next time". Lisa articulated:

I think the next pregnancy will not be as intense. [. . .]. I will not be able to absorb myself in a second pregnancy, with another child around. [Lisa; 6 months PN]

## Care

Care was strongly expressed during the two antenatal interviews, but not in the postnatal interviews. The first interviews showed a range of views that were categorised as "planning". At 22 weeks, women and their families start to think about the location in which they want to give birth. Birthplace locations include university hospitals, regional hospitals, birth centres and the woman's home. This is well illustrated by Katja, describing her thought process about choosing her place of birth:

We chose this birthplace by not looking at too many hospitals and birthing centres and I do not know what the other various possibilities are. You simply say "ok, this is coincidentally close by, and you go for it. . . or that the birth centre is actually the place you want to go and that you trust it. [Katja; 22 weeks AN]

Participants, planning to give birth at home or in birth centres, often reported receiving negative comments about these settings. Lana, who wished to give birth in a birth centre, recounts her conversation with her obstetrician:

She told me that I would die in a birth centre. She told me "you'll have a haemorrhage and then you have to act fast. . . ." I tried to laugh about it because I knew it was just her being stupid, trying to justify her job in the sense that she wants me to give birth in her hospital

because she makes money out of it. Anyway, it's her job and she believes in her job but still, she managed to scare me. [Lana; 22 weeks AN]

By the time of the second interviews this theme had fragmented itself into three distinct pathways. The first pathway merges with the *Decisions* theme "Negotiating the maze" which is outlined above. The second thread shows how either the initial choice is consolidated, or in some cases, participants were "Reacting to the unexpected". Bea, for example, disagreed with the idea of putting herself in the hands of others but nonetheless accepted that complications may occur:

I find that if you go with the idea of having a natural birth. . . if you have it in your head. I find a little unfortunate that maybe at the last minute they suggest a caesarean. On one hand I tell myself if it is the only solution because there is a risk for me or the baby at that moment I think I'll feel bad for a while. . .. especially after 16 or 24 hours of labour, I think I'll feel bad for a while. But on the other hand, as long as it's fine and it turns out well, the little bit of disappointment then passes. [Bea; 36 weeks AN]

This theme forks out and joins *Influences* at the "Mobilising resources" point, when women reach out to friends and family for support around the time of birth and early postpartum. Finally, this theme reappears and joins *Decisions* at the 6 months interview, when women talk about "Getting it right next time" by mentioning the type of care they would like for the next birth, as illustrated in Fig 2.

## Emotions

The theme of *Emotions* was noted during the first, third and fourth interviews but not during the second interviews. The first interviews revealed a "continuum of emotions", which reflected the intense expectations the participants had of their forthcoming birth. This means, women are reporting emotions that they had already felt prior to their pregnancy, and which they are expecting to experience until birth or beyond. For some, the main emotion was fear:

Fear of birth is already there, well fear of the pain. [Ronja; 22 weeks AN].

The fear of having to cope with a frightening experience can make women think about possible approaches to birth.

. . . so anyway, I understand one want to find an equilibrium. . . to have a birth that's as serene as possible, even if, in my opinion, it's going to be a battle field. . . especially the first time, when you don't know what to expect, so no matter how you represent it, how you imagine, no matter what you read or people tell you, it will be 10,000 times worse, 10,000 times different or 10,000 better than anything you can ever imagine or read. . . But it will be a battlefield. [Julia; 22 weeks AN]

Other representations of childbirth were more balanced. They relied on the participants' faith that childbirth is a natural process, traditionally achieved vaginally. These women tended to feel confident in their own capacity to give birth and cope with pain.

From my point of view, we are made for this, women, therefore, I do not see why there could be a problem or anything else. So, it's true, I am quite laid-back about that. [Lucie; 22 weeks AN]

Though absent during the second interviews at 36 weeks, *Emotions* was extremely strong during the third interviews, at 6 weeks postpartum. *Emotions*, rather than following a well-ordered continuum, were "polarised" ranging from strong frustrations to intense happiness.

. . .Luckily, she (baby) is cool anyway, because since the childbirth. . .. Well, I'm telling you about childbirth, I am well now. . . but I debriefed it, I debriefed it with Nathalie the second midwife, I was in tears, I wasn't well. . . I was not well, not well, I debriefed again with my gynaecologist, I spoke about it again with the hospital, it is clear that I never want to set foot there again. . .. [Leanne, 6 weeks PN]

In the final interviews *Emotions* were less to the fore but participants "Held on to powerful emotions". For Lisa this was a sense of disappointment:

It was a deep wish that I give birth naturally. A primal wish somehow. And then the disappointment that I did not make it. [Lisa; 6 months PN]

For most others, however, it was more positive. Olivia sums it up:

Simply magical, really how this child grows in your body and then somehow magically comes out and you just do not know how it was possible to be inside. And then after four months we started with baby purées but until then, I looked at him and thought 'Everything that he is, his existence somehow is because it went through my body'. And it is wondrous. Also, for my husband. And also, for a couple, to see that something like this is even possible. A gift really. [Olivia; 6 months PN]

## Influences

Women are influenced by internal and external factors. At the first interviews, this theme was labelled "Internal and external influences". Internal influences come from within and how women perceive and plan their journey to birth. Mina's main influences are internal in nature although she stressed the need for cooperation:

I like to be informed if I can. It calms me down. I have more control of the situation if I already know what is going to happen. Then, I can get more details as time goes on, but I would already have more or less of an idea of how things are progressing. This makes me feel calmer. I'm not wondering now how badly I'm going to do. I say to myself that I know these are the possibilities for the meantime, and then we will see. [Mina; 22 weeks AN]

Birth is a widely discussed topic, and society highly influences women and families. In Audrey's case, a wider world view was adopted. She spoke about society's influence on birth and her reaction to it:

I have the impression that this is what society sometimes makes us believe that it is possible when it is not. . . . . . and I think it has an influence on what people say when we hear women saying " anyway I do not want to give birth. I would like to be able to fall asleep completely and wake up when the baby is there" or things like that and I say to myself this is not how life and people in general work. [Audrey; 22 weeks AN]

Some women's journeys to their expected mode of birth can also be influenced by their growing baby. Leanne, who was particularly anxious about birth during the first interview, at

36 weeks articulated that her views have radically changed. She explained how her baby was the biggest influence:

> So childbirth is going to be easy with this state of mind. I find it a remarkable evolution since the last time... yes it's thanks to the baby because I felt it, because I saw its development and because it said to me: "I'm here... I'm a person and you managed to create me, now it's going to be okay, you'll see, I can do everything, I've already turned head down". [Leanne; 36 weeks AN]

"Internal and external influences," showed a continuous progression to the second interviews, around 36 weeks, when participants became more active in their birth preparations. As the birth approaches, women start planning and organising helping hands or other forms of support from friends or family members. At this time, *Influences* is referred to as "Mobilising resources". During her second interview, Isabelle spoke of how she "mobilised resources":

> I've got a very good relationship with my husband's family. They're really interested, and we telephone one another a lot. They also help me and I think after the birth they'll be really helpful. [Isabelle; 36 weeks AN]

Freya, however, worried about how she could be proactive.

> I worry about when I have to go into hospital because my partner might not be here because he's abroad. I do worry about what my role is, how I should prepare myself or is it all right. Will I have enough support from the people there? [Freya; 36 weeks AN]

As with the theme "D*ecisions*", the theme "*Influence*" was absent in the immediate postnatal period, but in this instance, re-emerged in the last interviews, at 6 months, when participants "Identified the most important influences" as they looked back at their births. Some women describe how their healthcare professionals were the greatest influencers:

> So the choice [for my birth] was somewhat influenced by the obstetrician. [Nina; 6 months PN]

> On the physical level I don't know [why it was so], but on an emotional level as I experienced it, was the information I had before from midwives; how they followed me, how they encouraged me, in my opinion. Although looking back, I never think about my obstetrician. He had such a tiny role in the whole thing, that I'm most grateful to the two midwives who helped in this positive experience. [Irene; 6 months PN]

Nora, who during pregnancy wanted a caesarean but gave birth spontaneously, saw the main influences as her baby and her obstetrician:

> I think it is my son and my doctor [who were the most influential]. The thing that convinced me was that it was safer for the baby, and that even if it was against my own wishes, I had chosen the safest path for the baby. [Nora; 6 months PN]

However, some women believe that women could merely be guided by external influences through their birth journey, but ultimately only internal influences matter:

Everyone has to find their own path and then be content with it. Also how you do things after the birth. There are so many people trying to tell you what to do. And at some point you have to build up self-confidence. I now know how I want to handle things with my child. Many might do things the same way but others might say: that is completely wrong. But as long as one has the feeling it is the right path for the child, I think it is the right way to go. And you have to learn that. [Wendy; 6 months PN]

Conversely, some women discuss their regret about their healthcare professionals' lack of influence, and how they would have preferred a stronger involvement in their birth journey's especially when an intervention was offered (informed consent):

I just recently realised in a conversation with a friend that because of the caesarean section I now have to wait for a year to get pregnant again. That upset me quite a bit. [. . .] I would have fought longer to try vaginal birth had I'd known. [Barbara; 6 months PN]

## Discussion

Qualitative studies by nature relies on language to obtain information from subjective experiences. Van Nes describes: "*The relation between subjective experience and language is a two-way process; language is used to express meaning, but the other way round language influences how meaning is constructed*" [38] (page 314). This rings even more true in Gadamerian hermeneutic studies. Hermeneutics of Gadamer seek to gain understandings through the spoken words. In Gadamerian studies, it is important to not only read transcripts, but to also read them whilst listening to the recording of the interview [31]. This hermeneutic study was carried out in Switzerland, and participants were interviewed in their own national language which were German, French or Italian. As per recommendations made by some researchers, the recorded data were transcribed in the original language and memos were written pertaining to significant codes prior to translating into English, thus maintaining as much as possible of the meaning behind the participants' words [38, 39].

The data we presented in the previous section show four hermeneutically derived themes. We developed them further in stage four of the research method, taking into account the pre-understandings each of us brought to the project, and our reflexive discussions throughout. Ratzinger identified a relevant hermeneutic of continuity and one of rupture [40]. Particularly noteworthy was that very few of the participants expressed a wish for caesarean section in the antenatal period but rather their expectations were on the need to have what they perceived as accurate information. Based upon that, they felt that they could make decisions that suited their own lifestyles preferences, although as shown in the final interviews, this did not always happen. Foremost in the mind of many participants was the health of their babies and if a caesarean was recommended for this reason they accepted it [15, 41]. Our findings support this schema but additionally include a hermeneutic of discontinuity which, rather than being disruptive, merely created a temporary gap in the hermeneutic of continuity. Such gaps have been described more vividly by some of the literature [42], Bergum, for example, suggested there is actually a rupture as a woman transforms into a mother during the birth process [43]. However, in this study, we have shown that the participants identified themselves as mothers before the birth of their babies as acknowledged by recent UNICEF report on the first 1000 days of life [44].

The absence of data in relation to *Decisions*, *Care* and *Influences*, that was revealed at the time of the third interviews, did not represent a complete break. This pause allows women to focus on getting to know their babies, to develop new routines, and to physically recover from the birth. The first weeks after birth are pivotal to ensure the flourishment of the mother-infant

bonding and attachment [45]. This has been described as a close emotional time when women often had little time or energy for anything apart from providing the necessary care for their babies [46]. It is now generally accepted that this is primarily due to the influence of various hormones [47, 48]. This resonated particularly in this study, as the participants, being first time mothers, were feeling their way through the early days of motherhood and getting to know their new baby. Yet, they were still reflecting on the birth of their baby when a sense of "coming of age" brought the themes to a turning point. By then, the themes developed into "Getting it right next time", "Holding onto powerful emotions" and "Identifying the most important influences". The expectation of education at this time was clearly shown, and this was particularly evident in the final interviews when participants reflected on the major influences during their pregnancies. However, it has been described as the time when their expectations were often unmet [49]. We speculate the birth outcome is determined by these influences because, as demonstrated by the data, the participants were happy to have given birth to healthy babies; a finding that is supported by other research [2, 25, 49].

Especially relevant is the discontinuity in the participants' expectations, as reflected in the second interviews in the theme of "*Emotions*". This may be due to mid to late pregnancy being a time when pregnant women's emotions have stabilised [48]. Additionally, our findings show that most of the participants' preparations for the birth are complete at this time and, while there was still a sense of fear expressed by some participants, there was general acceptance that things were liable to change and, in some ways, no more advance planning could be done.

Finally, we use the term "hermeneutic of rupture" to define the three trajectories that we have outlined in our themes of "*Care*" and "*Decisions*". Unlike Ratzinger who saw this phenomenon as negative, we found it to be more neutral as the themes themselves did not disappear totally but, after the first interview, became partially absorbed into two other themes. While some women initially expressed the desire for a caesarean section, most changed their minds during the course of their pregnancy. Nora's past experience of working in maternity services in developing countries for example, seems to have influenced her negatively on having a vaginal birth in relation to pain and complications. She did not trust herself and was also reassured by a precise date of birth. In the second interview, she still expected a caesarean but was contemplating the idea of vaginal birth, which she finally achieved and was happy with her decision. In the postpartum interview, however, these themes no longer existed. On reflection and further discussion, we believe this to be due to what women initially wanted to happen, balanced against acceptance of what actually happened. Our findings show this to be addressed by participants in other themes though not fully absorbed in them.

## Conclusion

This study sought to gain understandings around primigravid's expectations of birth, how they flourish during the antenatal period, and how they influence the lived experiences of the birth of their baby, given that Switzerland is known for its high CS rate and its medicalized approach towards birth. From this longitudinal study, four themes *Decisions*, *Care*, *Influences* and *Emotions* were recorded. The four themes and their intertwining paths during pregnancy and postpartum demonstrate the complexity of the expectations and experiences of women becoming mothers. *Planning* is strongly present during the antenatal period, and naturally disappears with the birth. *Emotions* is present mid pregnancy, is muted around 36 weeks pregnancy and holds a key part in women's narrative of their birth at 6 weeks and 6 months postpartum.

Primigravidas' expectations are greatly affected by internal and external influencers. External influencers may be healthcare professionals, friends and family, the media and society's

culture, whilst internal influencers are women's own beliefs and desires. *Influences* evolve throughout the longitudinal study period. Women's choice of birth mid pregnancy passively steers primigravid towards one choice or another, but by the end of the pregnancy, women actively seek and plan help and support for the time of birth and early postpartum days. After 6 months, women reflect on their journey and identify the most important influencers, often healthcare providers.

The sample in this study was "healthy primigravid women" as we were particularly concerned about the high caesarean section rate in the country. Several women did have emergency caesarean sections. Those that retained negative feelings at the final interviews, were not always those who had caesarean sections as satisfaction was more to do with the choices they made throughout the journey and control they were able to exercise. Therefore, women's experiences don't seem to be a strong factor contributing towards the rising caesarean section rates.

## Strengths and limitations

The aim of the study which was to develop a model of the emerging expectations of giving birth and the subsequent experiences of healthy primigravid women in four cantons in Switzerland, was achieved. It is the first study of its kind to be carried out in Switzerland. Even though Switzerland is a relatively small country, our findings may be transferrable to other countries in Europe and beyond due to its multicultural facets. Our findings derive from three major language regions in the country, each with its own culture and customs. This, however, brought unique challenges for this Gadamerian hermeneutic study with its emphasis on language. The decision to analyse entirely in English eased the process to providing a "common culture", but we also acknowledge that it is not the participant's culture, and some meaning behind words may have been lost in translation.

## Implications for further research and practice

With our focus on "healthy primigravid women" we add a dimension of new knowledge and provide a further layer to literature concerning the complex but under researched postnatal field. While our sampling strategy was intended to be as inclusive as possible, qualitative research can never be truly representative of the population. Thus, we plan to generate a questionnaire based on the findings, and once piloted and validated, administer this to a representative sample of first-time mothers in Switzerland.

In bringing together the data, the plan was not to compare the regions but to develop a model of the "Swiss" experience and the results of the analysis have focused on the commonalities. Nonetheless, it could also be of value in the future to consider the similarities and differences between the different regions of the country so that institutions such as insurers which cover the whole country can ensure they cover the most appropriate services.

The rising caesarean section rates seem to be related to factors other than women's preferences. Ambivalence towards a specific way of giving birth is common during pregnancy. This should be of concern for midwives and obstetricians during antenatal care. Information and counselling should be timely and comprehensive when discussing mode of birth. A negative birth experience may influence future preference for caesarean section. This should be considered by caregivers providing perinatal care.

Finally, since the study is limited to primigravid women, it would be interesting to see if multiparous women experience these stages in the same and at the same intensity. All drew on their experiences to learn from them and utilise them in their planning for future pregnancies. This has implications for health professionals who for almost a century have placed much of

their emphasis on antenatal care. It is worthy of consideration that, to make this more relevant, they assess the possibilities of providing a follow up visit to first time mothers at six months postpartum to enable the woman to reflect upon her birth journey.

## Supporting information

**S1 File. 4 E interview key questions.**
(DOCX)

**S2 File. 4 F interview key questions.**
(DOCX)

**S3 File. 4 D interview key questions.**
(DOCX)

**S4 File. 4 I interview key questions.**
(DOCX)

## Acknowledgments

We would like to thank all the women that participated in this study. The authors would like to thank Susanne van Gogh who developed the spiral model presented in this study. In addition, gratitude is extended to, Bénédicte Michoud and Laura Schirinzi for their part in the collection and analysis of the data.

## Author Contributions

**Conceptualization:** Valerie Fleming, Yvonne Meyer, Claire de Labrusse.

**Data curation:** Valerie Fleming, Franziska Frank, Yvonne Meyer, Piroska Zsindely, Claire de Labrusse.

**Formal analysis:** Valerie Fleming, Franziska Frank, Yvonne Meyer, Jessica Pehlke-Milde, Piroska Zsindely, Claire de Labrusse.

**Funding acquisition:** Valerie Fleming, Yvonne Meyer, Jessica Pehlke-Milde.

**Investigation:** Valerie Fleming, Franziska Frank, Yvonne Meyer, Piroska Zsindely, Claire de Labrusse.

**Methodology:** Valerie Fleming, Yvonne Meyer, Claire de Labrusse.

**Project administration:** Valerie Fleming, Yvonne Meyer, Claire de Labrusse.

**Resources:** Valerie Fleming, Yvonne Meyer, Jessica Pehlke-Milde.

**Software:** Valerie Fleming, Yvonne Meyer, Claire de Labrusse.

**Supervision:** Valerie Fleming, Yvonne Meyer, Claire de Labrusse.

**Validation:** Valerie Fleming, Yvonne Meyer, Claire de Labrusse.

**Visualization:** Valerie Fleming, Claire de Labrusse.

**Writing – original draft:** Valerie Fleming, Franziska Frank, Yvonne Meyer, Jessica Pehlke-Milde, Piroska Zsindely, Claire de Labrusse.

**Writing – review & editing:** Valerie Fleming, Franziska Frank, Yvonne Meyer, Jessica Pehlke-Milde, Piroska Zsindely, Harriet Thorn-Cole, Claire de Labrusse.

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
