## [Decision Letter · Decision Letter 0]

2 Nov 2021

PONE-D-21-14238Giving birth: a hermeneutic study of the expectations and experiences of healthy primigravid women in SwitzerlandPLOS ONE

Dear Dr. de Labrusse,

Thank you for submitting your manuscript to PLOS ONE. After careful consideration, we feel that it has merit but does not fully meet PLOS ONE’s publication criteria as it currently stands. Therefore, we invite you to submit a revised version of the manuscript that addresses the points raised during the review process.

We look forward to receiving your revised manuscript.

Kind regards,

David Desseauve, MD, MPH, PhD

Academic Editor

PLOS ONE

Journal Requirements:

3. Please include a copy of the interview guide used in the study, in both the original language and English, as Supporting Information, or include a citation if it has been published previously.

4. Please provide additional details regarding participant consent. In the ethics statement in the Methods and online submission information, please ensure that you have specified what type you obtained (for instance, written or verbal, and if verbal, how it was documented and witnessed). If your study included minors, state whether you obtained consent from parents or guardians. If the need for consent was waived by the ethics committee, please include this information.

6. Thank you for stating the following financial disclosure: 

7. Please note that in order to use the direct billing option the corresponding author must be affiliated with the chosen institute. Please either amend your manuscript to change the affiliation or corresponding author, or email us at plosone@plos.org with a request to remove this option.

8. PLOS requires an ORCID iD for the corresponding author in Editorial Manager on papers submitted after December 6th, 2016. Please ensure that you have an ORCID iD and that it is validated in Editorial Manager. To do this, go to ‘Update my Information’ (in the upper left-hand corner of the main menu), and click on the Fetch/Validate link next to the ORCID field. This will take you to the ORCID site and allow you to create a new iD or authenticate a pre-existing iD in Editorial Manager. Please see the following video for instructions on linking an ORCID iD to your Editorial Manager account: https://www.youtube.com/watch?v=_xcclfuvtxQ

9. Please upload a new copy of Figure 1 as the detail is not clear. Please follow the link for more information: " ext-link-type="uri" xlink:type="simple">https://blogs.plos.org/plos/2019/06/looking-good-tips-for-creating-your-plos-figures-graphics/"
" ext-link-type="uri" xlink:type="simple">https://blogs.plos.org/plos/2019/06/looking-good-tips-for-creating-your-plos-figures-graphics/"

Reviewers' comments:

Reviewer's Responses to Questions

**Comments to the Author**

1. Is the manuscript technically sound, and do the data support the conclusions?

Reviewer #1: Yes

Reviewer #2: Yes

2. Has the statistical analysis been performed appropriately and rigorously? 

Reviewer #1: Yes

Reviewer #2: N/A

3. Have the authors made all data underlying the findings in their manuscript fully available?

Reviewer #1: Yes

Reviewer #2: Yes

4. Is the manuscript presented in an intelligible fashion and written in standard English?

Reviewer #1: Yes

Reviewer #2: Yes

5. Review Comments to the Author

Reviewer #1: Understanding the reasons for increased rate of C-section is an important topic. Multiple points of view and type of reasearch may help. This paper point out the view of women before and after having giving Birth. Despite the diversity of culture in switzerland and a small number of patient, the authors found reasonably 4 mains themes to explore women's thougt patterns. These themes should help to sustain further studies and may be usefull to lead antenatal discussions.

I suggest to remove l152:"three gave Birth" wich is not usefull and suggests possible other interviewers biais (ethnicity, gender, nationality, age …).

Fig 1 appearence should be improve.

Reviewer #2: Dear Authors,

I really enjoyed reading your paper and its contribution ot the literature in this area.

these are just suggestions to enhance the review/background and further raise concerns re rising cesarean rates: perhaps include references to concerns as to the indication recorded as a justification for the CS i.e. if its fetal hypoxia is this corroborated in the notes. This was highlighted re maternal request in Brazil. Suggest also adding to concerns re the consequences of rising CS rates in terms of placenta accreta/percreta/increta and also the rising CS rates ineh USA accompanied by an increase in maternal deaths....

6. PLOS authors have the option to publish the peer review history of their article (what does this mean?). If published, this will include your full peer review and any attached files.

Reviewer #1: **Yes: **Alexandre Farin

Reviewer #2: No

---

## [Author Response · Author response to Decision Letter 0]

2 Dec 2021

Dear Dr. David Desseauve,

We would like to thank you for considering our manuscript, and the reviewers for their time and comments. We have edited our manuscript and supporting information to address the raised concerns:

The style of the manuscript has been changed to meet the PLOS One requirements.

2. Please review your reference list to ensure that it is complete and correct. If you have cited papers that have been retracted, please include the rationale for doing so in the manuscript text, or remove these references and replace them with relevant current references. Any changes to the reference list should be mentioned in the rebuttal letter that accompanies your revised manuscript. If you need to cite a retracted article, indicate the article’s retracted status in the References list and also include a citation and full reference for the retraction notice.-------------------------------------------------The following references have been added following the comments of reviewer 2:

 o 11: Morton R, Burton AE, Kumar P, Hyett JA, Phipps H, McGeechan K, et al. Cesarean delivery: Trend in indications over three decades within a major city hospital network. Acta Obstetricia et Gynecologica Scandinavica. 2020;99: 909–916. doi:10.1111/aogs.13816

 o 13: Sobhy S, Arroyo-Manzano D, Murugesu N, Karthikeyan G, Kumar V, Kaur I, et al. Maternal and perinatal mortality and complications associated with caesarean section in low-income and middle-income countries: a systematic review and meta-analysis. The Lancet. 2019;393: 1973–1982. doi:10.1016/S0140-6736(18)32386-9

3. Please include a copy of the interview guide used in the study, in both the original language and English, as Supporting Information, or include a citation if it has been published previously.------------------The interview guides in English, French, German and Italian are now included. 

4. Please provide additional details regarding participant consent. In the ethics statement in the Methods and online submission information, please ensure that you have specified what type you obtained (for instance, written or verbal, and if verbal, how it was documented and witnessed). If your study included minors, state whether you obtained consent from parents or guardians. If the need for consent was waived by the ethics committee, please include this information.---------------------------------------------4. We have specified in our manuscript that women were asked to sign a consent form.

When you resubmit, please ensure that you provide the correct grant numbers for the awards you received for your study in the ‘Funding Information’ section. ------------------------------------- When submitting, we will ensure the funding information matches between the manuscript and the financial disclosure.

6. Thank you for stating the following financial disclosure: 

Please include your amended statements within your cover letter; we will change the online submission form on your behalf.------------------------------------------------------We have stated that the funders had no role in study design, data collection and analysis, decision to publish or preparation of the manuscript

7. Please note that in order to use the direct billing option the corresponding author must be affiliated with the chosen institute. Please either amend your manuscript to change the affiliation or corresponding author, or email us at plosone@plos.org with a request to remove this option.-------------------------------------------- Please, would you mind changing the corresponding author to Valerie Fleming of the Liverpool John Moores University. 

8. PLOS requires an ORCID iD for the corresponding author in Editorial Manager on papers submitted after December 6th, 2016. Please ensure that you have an ORCID iD and that it is validated in Editorial Manager. To do this, go to ‘Update my Information’ (in the upper left-hand corner of the main menu), and click on the Fetch/Validate link next to the ORCID field. This will take you to the ORCID site and allow you to create a new iD or authenticate a pre-existing iD in Editorial Manager. Please see the following video for instructions on linking an ORCID iD to your Editorial Manager account: https://www.youtube.com/watch?v=_xcclfuvtxQ---------------------------------------- Prof. Valerie Fleming’s ORCID number is 0000-0002-4672-4843

9. Please upload a new copy of Figure 1 as the detail is not clear. Please follow the link for more information: https://blogs.plos.org/plos/2019/06/looking-good-tips-for-creating-your-plos-figures-graphics/" https://blogs.plos.org/plos/2019/06/looking-good-tips-for-creating-your-plos-figures-graphics/"---------------------------- Thank you for pointing this out. We have now improved the quality of Figure 1, and checked it meets the journal's specification on PACE.

Reviewer 1 comments:

Understanding the reasons for increased rate of C-section is an important topic. Multiple points of view and type of reasearch may help. This paper point out the view of women before and after having giving Birth. Despite the diversity of culture in switzerland and a small number of patient, the authors found reasonably 4 mains themes to explore women's thougt patterns. These themes should help to sustain further studies and may be usefull to lead antenatal discussions.

I suggest to remove l152:"three gave Birth" wich is not usefull and suggests possible other interviewers biais (ethnicity, gender, nationality, age …).

Fig 1 appearence should be improve. -------------------------------------------------------------------------------- We would like to thank reviewer 1's positive and constructive feedback. We agree with the made comments and removed l172: "three gave birth", and as above (see item 9) improved figure 1.

Reviewer 2 comments:

I really enjoyed reading your paper and its contribution ot the literature in this area.

these are just suggestions to enhance the review/background and further raise concerns re rising cesarean rates: perhaps include references to concerns as to the indication recorded as a justification for the CS i.e. if its fetal hypoxia is this corroborated in the notes. This was highlighted re maternal request in Brazil. Suggest also adding to concerns re the consequences of rising CS rates in terms of placenta accreta/percreta/increta and also the rising CS rates ineh USA accompanied by an increase in maternal deaths.... ----------------------------------------------------------- We would like to thank reviewer 2's suggestions and positive feedback. Reviewer 2 has raised to interesting points which we have addressed: 

- L66-67: a sentence was added reporting the common medical indications for CS. 

- L.70-71 : another sentence was added reporting the concerns re the rising CS rates. 

2 References were then added. See point 1.

We would like to thank you for giving us the opportunity to enhance our manuscript with your constructive comments. We have made sure that we have incorporated your feedback and hope that our manuscript now meets the PLOSone standards and convinces you to accept it for publication. We look forward to hearing from you regarding our submission. We would be happy to answer any further queries and comments that you may have.

Best wishes

Claire de Labrusse, on behalf of Valerie Fleming

Corresponding Author: Valerie Fleming

Liverpool John Moores University, Faculty of Health

Rodney House, Mount Pleasant, Liverpool L3 5UX, United Kingdom

V.Fleming@ljmu.ac.uk

[Tel: 0044 (0) 151 231 4017]

Additional Contact [should the corresponding author not be available]

HESAV

School of Midwifery

Av. de Beaumont 21, 1011 Lausanne, Switzerland

claire.delabrusse@hesav.ch

[Tel: 0041 (0) 21 316 81 60 ]

---

## [Editor Report · Decision Letter 1]

14 Dec 2021

Giving birth: a hermeneutic study of the expectations and experiences of healthy primigravid women in Switzerland

PONE-D-21-14238R1

Dear Dr. Fleming,

We’re pleased to inform you that your manuscript has been judged scientifically suitable for publication and will be formally accepted for publication once it meets all outstanding technical requirements.

Kind regards,

David Desseauve, MD, MPH, PhD

Academic Editor

PLOS ONE

---

## [Editor Report · Acceptance letter]

27 Jan 2022

PONE-D-21-14238R1 

Giving birth: a hermeneutic study of the expectations and experiences of healthy primigravid women in Switzerland. 

Dear Dr. Fleming:

I'm pleased to inform you that your manuscript has been deemed suitable for publication in PLOS ONE. Congratulations! Your manuscript is now with our production department. 

Kind regards, 

on behalf of

Dr. David Desseauve 

Academic Editor

PLOS ONE